# Construction of Teacher Professional Identity through Initial Training

Marcela Casanova-Fernández [1,*], Jorge Joo-Nagata [2] , Emily Dobbs-Díaz [3] and Tricia Mardones-Nichi [4]

1 Doctoral Program in Education, Metropolitan University of Educational Sciences, Santiago 776019, Chile
2 Nucleus Computational Thinking and Education for Sustainable Development (NuCES), Center for Research in Education (CIE-UMCE), Metropolitan University of Educational Sciences, Santiago 776019, Chile
3 Department of Pedagogical Training, Metropolitan University of Educational Sciences, Santiago 776019, Chile
4 Department of Basic Education, Metropolitan University of Educational Sciences, Santiago 776019, Chile
* Correspondence: marcela.casanova2016@umce.cl

**Abstract:** The construction of a professional identity is a key element in the transition to the teaching profession, which begins its development at the initial training, especially during professional training courses. Purpose: This article collects the experiences related to the construction of the teaching identity of teachers who graduated from two state universities of the Santiago Metropolitan Region in Chile. Methodology: Interpretative-qualitative research was carried out, framed in a phenomenological design. Semi-structured interviews were applied to 10 teachers who graduated from a course of Primary Education Teaching in 2014 and are still practicing the profession in the school system. Categories and subcategories were created from their tellings, establishing relationships between teacher identity and initial training. Results: This study shows that teachers value their initial training, especially regarding values, discipline, and experiences of training in different contexts as constitutive elements of their professional identity. However, both groups describe unsatisfactory situations experienced as beginner teachers, where they recognized the weaknesses of their initial training for overcoming the complexities of the school context. Conclusions: This study points to the need to strengthen the articulation in the process of training from university onwards and the support of the educational establishment.

**Keywords:** professional teaching identity; teacher training; professional training; beginner teachers

## 1. Introduction

Research on Initial Teacher Training (ITT) acquires relevance for the promotion of the generation of strengthening higher education policies for the construction of the professional identity of teachers, providing knowledge that allows to guide and direct training that is consistent with the requirements and demands of the current education [1]. The literature states that to respond to the requirements of the twenty-first century, a solid initial teacher training must be provided. This must respond to the job's complexity and be reflected in the competencies, performances, and skills of future teachers [2].

In this context, Law 20,903 on Teacher Professional Development, Law 20,129 on Quality Assurance of Higher Education, and the Guiding Standards in Law 20,370 on General Education emerge in Chile. These lead the formative trajectory from the beginning of ITT through situated, community learning experiences and are generators of applied knowledge that allow for the construction and strengthening of professional identity [3–5].

In summary, it has been said that professional identity begins from ITT, strengthening, nurturing, and evolving throughout professional life [6]. This identity process is presented as thoughtful, dynamic, and interactive. It is linked to the deep knowledge of the discipline content, to the beliefs of how it is taught and how it is learned, to the emotions, and to the quality of the experiences of the professional action [7]. Therefore, the construction of the teaching identity is the result of the multiple aspects that are associated with personal and

professional life experiences. Likewise, they are driven from the ITT and are constructed and reconstructed throughout the professional exercise [8]. The first approach to the profession and the recognition of teachers themselves and the community occurs in professional training courses. These training courses are guided, advised, and formalized from ITT, to become spaces for reflection on the profession and teachers themselves [9,10].

However, studies show that the first crises of professional identity begin in the situated experiences of professional training courses. In this context, the tension between the concept of the ideal teacher and the real teacher in the daily activities of the classroom and the educational spaces of the school comes up. In other words, tension occurs in the educational action, which is situated in the practice itself [11]. This tension is further exacerbated at the beginning of the professional exercise asof beginner teachers. These identity crises must be conducted, from the beginning, in directed instances of teacher reflection that cover topics about the profession, based on empirical evidence. This way, it would be possible to know and understand the elements of the work environment that will be faced and provide challenges throughout a professional career [12,13]. In addition, in this way, the loss of a work purpose and the lack of an analytical habit that leads to a standstill or desertion could be avoided [14,15]. All this could have a direct impact on the effectiveness and equity of education [16].

The evidence referring to the crises in the process, trajectory, and professional development of teachers raises the need for having a body of theoretical knowledge that guides the construction and strengthening of the identity of teachers [17]. For this reason, qualitative research was carried out to reduce the theoretical gap related to the scarce study of practical training in Chile within the initial training [18]. In this regard, the research collected opinions on the construction of the professional identity of teachers of Primary Education Teaching who graduated from two universities of the Santiago Metropolitan Region in Chile.

## 2. Theoretical References

### 2.1. Professional Identity

The construct of professional identity becomes complex when it comes to a definition, given its dynamic nature, which also depends on the sociohistorical context and belongs to a personal dimension. This last dimension is constantly constructed and reconstructed through the professional trajectory in social interactions and individual reflections [19]. Thus, it is understood that the professional identity of teachers, as well as the identities of other professions, will be the result of both individual and collective constructions. This will allow for the identification of different groups of professionals [20]. The specificity of a professional identity will be based on its purpose and the context in which it is developed. However, professional identity is also part of a complex process of social recognition, as teachers are recognized by others, and due to social relationships, which connect both members of the professional group and other groups [21]. It is also interpersonal, since it implies self-recognition as a determined professional and is recognized as such within a particular context, thus generating a specificity in the relationship of identification and differentiation [22].

The development of the professional identity of teachers is understood as the construction that is carried out throughout the life of a teacher. It is a continuous process of the idea and the image that teachers construct for themselves, in terms of teaching and learning, the roles they play, and their professional and ethical objectives and purposes. All this is in a macro dimension made up of affective and cognitive, personal, and social dimensions, which is constantly projected to the past and the future, mediated by policies that define the standards of their profession [23]. Concerning this idea, the construction of the professional identity of teachers will be linked to multiple factors that converge in a sociocultural space and that determine the projection of teaching work. In this context, historical moments that have led to generating a mark on the teaching identity are recognized, which starts from the viewpoint of the apostolate, remaining until now with the concept of vocation,

passing through the concepts of the technicians of the profession and the generators of knowledge [24].

The external demands of today's society, with their innumerable demands on the teaching task, and the changes of students and their ways of connecting have led to a crisis in the professional identity. This makes it complex to predict the future formative necessities of teachers [25]. In this scenario, teachers must respond to administrative-type tasks that the operation requires in addition to complying with the national curriculum and with standards of international measurements, and provide emotional support to students and their families [26]. Thus, they are considered responsible for everything happening in the classroom, with scarce opportunities to reflect on their own practices. This implies a questioning of the quality of their own work [27]. In short, this leads to a generalized discomfort, which is expressed in doubts, resistance, disappointment, and the sensation of emotional exhaustion [28]. This tension between the role of the ideal and real teacher leads to burnout, triggering the loss of work purpose and progressive autonomy, loneliness, and isolation, with a consequence of the de-professionalization and abandonment of the school system [29].

*2.2. Initial Teacher Training*

In the process of shaping the professional identity of teachers, different factors come together from social, cultural, and school contexts, from the stages of development of the teaching career, and from interactions with members of the educational community. This configuration occurs in a continuum that begins intentionally in initial training, specifically in professional training courses, which constitute an important route for the conformation of the teaching identity [30]. Therefore, during recent decades, both nationally and internationally, interest has been placed on the generation of policies strengthening Initial Teacher Training (ITT), since the existence of a direct relationship between teaching quality and student performance has been demonstrated [31,32]. Between 1997 and 2002, the Strengthening Teacher Training Program (FFID, by its initials in Spanish) was implemented, with the purpose of improving the academic quality of students in pedagogical careers, increasing the number of students and entry scores, and increasing the retention rate [33].

Currently, the laws on the Professional Development System –Law 20,903– [34], Quality Assurance –Law 20,529– [35], School Inclusion, and New Public Education have given guidelines to guide the path and mechanisms to strengthen ITT programs, in response to the challenge of integrating disciplinary knowledge, pedagogical knowledge, experiences, and knowledge of practice, in conjunction with reflection [36]. ITT is then constituted as a relevant and complex space. It affects the construction of the professional identity of the future teacher through the training opportunities that are provided, especially in professional training courses [37]. In these training courses, the acquisition of skills is acquired to link and articulate theory with praxis, teaching, and learning [38,39]. The development of critical positions that allow for addressing the complexity of the educational act in diverse and inclusive scenarios [40] and the ability to "manage stressful, challenging and emotionally exhausting situations" are also acquired [41] (p. 30).

Following the above, the period of professional training during ITT undoubtedly constitutes an essential element in the training and performance of future teachers in the school system. However, numerous studies have reported the disarticulation and fragmentation between theory, practice, and pedagogical reflection, generating tensions between initial training policies and the academic results of students [42,43]. On the other hand, the Report of the Commission on Initial Teacher Training [44] points out, as one of its critical points, the lack of the conceptualization of pedagogical knowledge as the articulating axis of training processes; thus, the relevance of the progression of training courses as an approach to professional life is highlighted, based on the systematic reflection around the knowledge of skills, know-how, and being a teacher, which are elements that arise from the teaching and learning processes experienced in the systemic and situated contexts where pedagogical knowledge begins [45].

## 3. Materials and Methods

### 3.1. Research Focus

The methodology used in this study is part of the qualitative interpretative paradigm with a phenomenological research design [46], since the main interest was to understand the meaning of the experiences in the trajectory of teachers of Primary Education Teaching, who reflect on the construction of their professional identity and their initial training.

### 3.2. Study Context

For this study, two Chilean state universities of the Santiago Metropolitan Region that teach Primary Education Teaching were considered. These universities were selected according to criteria of convenience, including accessibility and proximity to the field of study. Regarding accessibility, a letter was sent to the Primary Education Teaching career directors of both Schools of Education, to request access to the field of study through the databases of teachers of primary education who are working as teachers in the school system. Informed consent was provided stating the voluntary and disinterested nature of participation as well as the protection of data and confidentiality.

### 3.3. Participants

A database was prepared with the information provided, and teachers were contacted via email to explain the purpose of the study and the selection criteria. Subsequently, telephone contact was made with those teachers who agreed to participate. Informed consent was sent via email and the date, time, and place of the interview in person were coordinated. The participants were distributed according to the information indicated in Table 1.

**Table 1.** Distribution of participants.

| Participants | University | Sex |
|---|---|---|
| Participant 1 | Universidad de Santiago de Chile (USACH) | Female |
| Participant 2 | Universidad de Santiago de Chile (USACH) | Female |
| Participant 3 | Universidad de Santiago de Chile (USACH) | Male |
| Participant 4 | Universidad Metropolitana de Ciencias de la Educación (UMCE) | Female |
| Participant 5 | Universidad Metropolitana de Ciencias de la Educación (UMCE) | Male |
| Participant 6 | Universidad Metropolitana de Ciencias de la Educación (UMCE) | Female |
| Participant 7 | Universidad de Santiago de Chile (USACH) | Female |
| Participant 8 | Universidad Metropolitana de Ciencias de la Educación (UMCE) | Female |
| Participant 9 | Universidad de Santiago de Chile (USACH) | Female |
| Participant 10 | Universidad Metropolitana de Ciencias de la Educación (UMCE) | Male |

As shown in Table 1, the number of participants was 10: 5 graduates from the Universidad Metropolitana de Ciencias de la Educación and 5 graduates from the Universidad de Santiago de Chile. The gender distribution was 7 women and 3 men.

### 3.4. Techniques Used

The collection of information was carried out through a semi-structured interview, which is a technique that allows participants to interpret the phenomena of their teaching experiences from the ITT for the construction of their professional identity. A script was prepared before the interview with questions that addressed the study topics concerning the objectives and establishing guiding questions, which are described in Table 2.

The semi-structured interview was applied in person to each participant, according to their availability of time and where they considered most appropriate. Before starting the interview, an informal conversation was held to thank them for their willingness to participate and to generate a pleasant atmosphere that allowed for a fluid conversation. At the beginning of the interview, the informed consent was read, and authorization to record

the audio of the interview was requested. Each interview lasted approximately 60 min, and the audio recordings were transcribed verbatim.

**Table 2.** Semi-structured interview script.

| Topics | Guiding Questions |
|---|---|
| Construction of Professional Identity | What does it mean for you to be a teacher? |
| | How would you define your role as a teacher? |
| | What does it mean for others to be a teacher? |
| Initial Teacher Training | What experiences from your ITT have been relevant or meaningful to your professional performance? |
| | What elements do you think were missing from your initial training? |
| | What experiences do you identify from your professional training courses that contributed to the construction of your professional identity? |

Source: Prepared by the author.

### 3.5. Data Analysis

The methodology for the analysis was inductive using the Constant Comparative Method [47], which comprises the following phases:

1. Comparison of applicable events for each theory: The initial task consisted of coding each event and comparing it with other groups of events, similar or different, coded with the same category. This started with a coding, called initial or open [48]. This first part was intended to maintain closeness with the information collected, avoiding preconceived categories, so the codification that was carried out emanated from the action, discourse, or intention that was in the data [48]. At this stage, the search for new information was systematically achieved until theoretical saturation was reached.

2. Integration of the categories and their properties: According to Charmaz [48], the second step was focused coding, which consisted of the selection of the codes that had greater importance and/or those that emerged more frequently in the data (what prevailed in this coding is that the selection had the greatest possible coherence and, at the same time, approached the studied research problem). Then, the axial coding stage was reached, where the different codings were related; thus, it was possible to create the relationship between categories and subcategories. This phase contributed to the coherence and detailed analysis of the two previous coding phases. The analysis consisted of systematically comparing events and categories to achieve a greater approach to the phenomenon of study, modifying the properties of the categories in the initial comparisons and integrating these with other categories of analysis. The entire categorical analysis process was carried out with the support of Atlas.TI v 8.0 software (Frankfurt, Germany).

### 3.6. Research Ethics Codes

The criteria that guided this research were the search for truth and honesty, and the results presented were those that were obtained in the investigation process without any distortions. In addition, the criteria resulted in concern about the treatment of participants through clarity in their intentionality. In this regard, we thought to investigate the construction of professional identity and the experiences of the initial training of the teachers who remain in the school system; the use of informed consent that included the information sheet and the consent form were submitted for discussion by and approval from the research ethics committee. On the other hand, concerning the originality of the work, the authorship was the researchers', and the real data that emanated from the participating subjects was used [49]. In addition, this research had as an ethical requirement the criteria of scientific rigor, which included accepted methods with appropriate analysis techniques to produce reliable data [50].

A relationship of mutual trust and professional integrity between researchers and participants was favored, who, in this case, were teachers that remained in the school system. They also actively participated in the study, as their life lessons and experiences were fundamental to the research. However, investigators assured that the information would be safeguarded and that its confidentiality would be protected.

Teachers who participated in the research had the right to withdraw and revoke consent at any time without giving reasons, yet none of them withdrew from participation.

## 4. Results

Based on the above and the objectives of the research carried out, the results by category and subcategory are presented below, to divulge the process of construction of teaching identity from the teacher's voices.

### *4.1. Construction of Professional Identity*

This category is characterized by the ability of the beginner teacher to construct their identity as a teacher, what it means for them to be a teacher, how they define being a teacher, and the meaning of being a teacher for others.

#### 4.1.1. Meaning of Being a Teacher

The subcategory Meaning of Being a Teacher reveals the ethical value of being a teacher. Teachers highlight those values that have led them to overcome the obstacles of the first years of employment.

- Teachers stay because they want something, to fight for something. [Pp1]
- For me, being a teacher means responsibility, with what I do, and with what I say. [Pp1]
- Being a teacher has to do with the word support. [Pp6]
- For me, being a teacher means to be, to have a goal, to defend it, and to be always the same. [Pp1]
- When you are a teacher, you know and start studying knowing that the money is never going to be good, and then you study, I think, as the majority may think, that most of us study for vocation, because we like it. [Pp10]

The commitment to the profession regarding academic preparation and the role it plays with students are also highlighted, and examples of what should and should not be done for academic preparation are mentioned:

- The teacher also must be a mediator of knowledge. [Pp3]
- Of course, you need to be very responsible for that, that is, knowing that if I don't prepare for a class and I arrive and just do a book activity, it's disrespectful. [Pp2]
- Teachers must empower themselves to be an entity of change, a transformative entity, a researcher who make knowledge for their peers, for the scientific community. [Pp10]

#### 4.1.2. Construction of their Teaching Being

The subcategory Construction of their Teaching Being considers the tellings about the process that teachers point out as important for the configuration of their professional identity. Among them, disciplinary learning stands out:

- It is constantly being constructed from the study too and that is why I feel the need to continue studying. [Pp1]
- Difficult experiences are what they tell you 'OK, this is what needs to be improved, here I have to continue this way, in this I have to perfect myself because here is where I am messing up'. [Pp2]
- My idea was always to improve and keep improving to change what was happening there; that was my idea, even after I started my studies. [Pp10]

On the other hand, the teachers point to peer learning as an important element for the construction of their teaching being:

- I used to watch, I have always been like this, super observant, I watched what my companions did, I watched and watched. [Pp4]
- It was that support, of being able to make a mistake, of helping you, of having the right word at the right time, of accompanying you, almost like a mentorship. [Pp7]

### 4.1.3. Meaning of Teaching for Others

This subcategory responds to the perception that teachers have about what the profession means for society, and the value it gives:

- We are technicians of pedagogy, we replicate, we are provided with bases already prepared, we replicate that knowledge, year after year, we plan, but we never get out of that limit. [Pp10]
- Schools continually cauterize teachers, they shape teachers into their operating box, and despite teachers arrive with many new practices, original practices, or with a lot of sense of vocation, the system slowly consumes them. [Pp5]
- Society has gotten used to seeing the school as a nursery school and not as a training agent, minimizing the work of the teacher. [Pp5]

### 4.2. *Initial Teacher Training*

This category refers to all those experiences lived by teachers in the initial training stage, which guided the construction of the teaching being. Two subcategories, undergraduate disciplines and professional training courses, have been identified from the teachers' stories.

### 4.2.1. Undergraduate Disciplines

This subcategory refers to the experiences in the courses taken during Initial Teacher Training, and the impact these had on the construction of their professional identity. It shows the negative experiences experienced by the teachers, as they express a critique of the institution's weaknesses in their undergraduate training, specifically topics of the profession such as the relationship with guardians and parents:

- No, in university parents were never mentioned, everything was with the student, in university we had nothing to do with their parents. [Pp8]
- Give more lessons, as I say, related to the work with the guardians, which is the weakest thing. In initial training we did have a course that was lead teacher and orientation; but maybe it would be good to include a semester with a course related to the work with parents in schools, lead teacher of a class and even voice impostation. [Pp7]

Another topic that they consider weak in initial training refers to the emotional tools to stay in the profession:

- I believe that the initial teacher training does not provide many tools to survive in the teaching field, it provides the knowledge maybe disciplinary and pedagogical, but it does not provide many tools to support ourselves. [Pp10]
- I wish they had taught me certain tools, and that I could be more assertive and establish better bonds. [Pp1]

Some teachers identified specific disciplinary aspects:

- Regarding the evaluation, there were many shortcomings within the university; 2011 was the first year of programmed evaluation, but, due to a student strike, there was no evaluation. [Pp5]
- The theme of my thesis, special educational needs; no one taught us to work with that. [Pp8]
- I would have appreciated it if we have been provided with more strategies to work with behaviorally conflicted children and with children with learning problems because I also feel that our program lacks a lot of these topics. [Pp6]

-   The program missed some updates too, because I remember that when we had the ICT course, an interactive board was presented to us, but I never saw that very old board later, it was an interactive board that no longer exists. [Pp6]

Among the aspects indicated, administrative issues that were not taught in the initial training but that are key to job performance were mentioned:

-   Something super simple like filling out the class register and the grade book, administrative issues like that, because when you arrive at the school for the first time, you discover that the books are sacred, then you do not know what to do with the books when you are there and makes you feel as if adrift; there are several things that the initial teacher training does not cover. [Pp10]
-   You studied directly pedagogical matters [...] because I wanted to, not because of my training courses -hey, I don't know, nobody said 'let's go to rehearse how to fill a book, where to sign and what the lectionary and activity refer to- '. [Pp2]
-   I did not know what my duties and rights were concerning being an education worker, nor did we manage the labor code, we never studied it in the university, and I think it is an important point to be able to demand certain basic conditions so that you can develop in the profession. [Pp1]

Some teachers refer to research and skills regarding the development of these:

-   You are trained to be a teacher, that's all, you graduate, and you have to replicate everything that the ministry curricularly indicates and such, but they do not tell you: 'no, it's just that the teacher has to be a researcher too', 'the teacher has to be an information collector as well'. They do not teach that. [Pp10]
-   Lack of things such as 'research the basics, research this, research the policies, what kind of didactics you're going to use', 'look, this is the context'. I believe it needs to be much more practical in that sense. [Pp1]

On the other hand, it is possible to identify stories of teachers who relate situations that are specific to their higher education institutions, in terms of organization that impacted their vocational training negatively:

-   We were like an experiment because we were in the first year in 2010, where there were many changes and many modifications, in fact, I think there are courses that are no longer taught and teachers who are no longer there, like it was a bit of experimenting with us. [Pp6]
-   I felt that it was not enough, that I did not receive enough classes, but it is because we had many gaps due to the strikes in the university. [Pp6]
-   On the way, we designed our modules and the curriculum. We even were designing the degree process regulations. [Pp5]

As for the positive experiences lived in the undergraduate stage, which strengthened professional identity, these are related to the professors that they interacted with:

-   There were professors who, from the moment we started to have classes with them, made us change that perspective, something as simple as the belief that the content is everything, and there were professors who took the risk and said 'Forget about the content, what you need to know is how to get there, the content is secondary, so to speak, what you are interested in is working on the skill'. [Pp5]
-   For example, science classes with the same professor NN were difficult, but he taught us different strategies and methodologies for us to be able to teach. [Pp9]
-   They worked a lot with project methodology, which is something that I can now apply in terms of scientific research projects. [Pp5]
-   The way of studying with professors was always based on the idea that there is no answer, they do not come to teach us a panacea in education, but they do come to problematize, to generate a problem to which we must be able to generate solutions or strategies or a project, whatever needed to solve that problem. [Pp5]

### 4.2.2. Professional Training

Here, reference is made to the situations experienced in the different aspects of professional training studied in Initial Teacher Training, and the impact these had on the construction of their professional identity, as well as to the support of the advising teacher at the school. Positive experiences that the teachers lived through, with the support of the advising teacher, are reported:

-   Nothing to say about the training courses, my professor, also my professor of training observation at the university gave us the training classes already prepared and said: 'Go! it's your turn at this school'. And it's super good that they make it easy for you, so you can do your training. [Pp6]
-   That's a nice process that has at least the professional career. [Pp5]
-   I had a hard time and my advisor professor in the university supported me a lot in the process. [Pp5]

The diversity of training centers also emerges as a positive experience, through the validation of heterogeneity and knowledge of different contexts for decision-making when looking for a job:

-   Most of the training classes assigned to me were in more vulnerable schools which was wonderful for me because I could see the reality and discover what I was really interested in. [Pp4]
-   Do you know why I find it good that the training classes are as diverse as they are? Because you come out of the bubble where you are as your reality, it is like a social shock, it's intense. [Pp6]
-   It is having the vision of totally different schools, for example, a municipal school of Pudahuel compared to a private school in Las Condes; then I wanted to see that, different things. [Pp7]
-   We had a complex school where I had a good time, paradoxically, I had a good time because I learned to read the children and love them in their context. [Pp5]
-   When I started to see those realities, I started to open my eyes a little bit and realized what kind of school I wanted to be in. [Pp6]

The support of the advising teacher of the educational establishment has both positive and negative tellings. In the positive area, teachers point out the importance of the guidance received:

-   I had a very good advising teacher who warned me from the first day, she was the coordinator too and kind of talked to me and told me that we were in a complicated situation, that there had been some accusations against a teacher inside the school a few years ago. So, the guardians and parents were very picky, and she gave me a lot of tips on how to handle that situation. [Pp5]
-   The teacher that I had in my professional training course treated the children with a lot of differences. [Pp4]

As for the negative experiences, aspects related to the treatment received by the advising teachers are highlighted:

-   My teacher was also very punitive with her comments, in what she said; one training in special, which was my last training, and in a school that was also Catholic, in fact almost made me abandon the career. [Pp1]
-   I had a bad advising teacher, who was not a teacher, she was a pharmaceutical chemist and, for one of those things that just happen, she had to do classes; she had poorly teaching elements and questioned a lot about what I did. [Pp5]
-   Initially, in a training session, I said: never again, I don't want this. [Pp8]
-   A teacher scolded a child and the child said: Are you sure? Let's see if you will be seated here tomorrow. Just like that, and the teacher was not even able to admonish that child. [Pp8]
-   In training courses, teachers see you as a menace. [Pp5]

Finally, it is possible to highlight the tellings of teachers that refer to what they would have liked to include in their professional training courses, as recommendations they have from their lived experience:

- I would have liked to be given the possibility of doing more within my training, meaning, not only being assigned to certain tasks or being told 'go over there and do all this stuff'. I wish they would have also let me some field of action. [Pp1]
- I would have liked something that would have helped me to get out of that student field, how to be more of a leader, more independent. [Pp1]
- I believe that, if such mentoring existed, it should be, in the case of training courses, well linked with the institution; because it is also necessary to understand that the university is a bubble, a total bubble. [Pp3]
- I think the training courses should be more meaningful and give us more importance because I felt I was always like the assistant. [Pp8]
- Having more support because I remember that my professors never went to my training classes, except for one training where a professor worked precisely in that school and it was the only training that I had where there were always people, but in the other cases, we were always alone. [Pp8]
- To approach or make this transition to the educational reality, the mentoring of the advising teacher of the training should be complemented with the mentoring of some other people like, I do not know, the inspector general of the school or some other authority related to the disciplinary areas and with the direct relationship with teachers. [Pp3]

## 5. Discussion

From the results of the analysis, different categories emerge; the first category accounts for the Construction of Teacher Identity, emerging with three subcategories: the Meaning of Being a Teacher, the Construction of their Teaching Being, and the Meaning of Being a Teacher to others. For the first two, the ethical component of the profession stands out, in relation to the commitment and social responsibility that is required for exercising it. This idea arises from the same teachers in their situated pedagogical action. Likewise, awareness of the need for permanent disciplinary preparation and updating arises. However, in the third subcategory, the tellings show situations of the devaluation of the profession by society and the educational system itself. This coincides with what was pointed out by Ávalos and Sotomayor [51], regarding the identification of the five sources of conflict related to the teaching identity, which include questioning the quality of teaching work and doubts about its legitimacy (these affect its value).

As for the Initial Teacher Training category, the Undergraduate Disciplines and Professional Training subcategories emerge. In both subcategories, teachers identify both positive and negative experiences that influenced their early years of professional performance. They point out that neither of the two universities provided adequate preparation for the acquisition of the emotional tools needed to be able to solve the complex situations of their profession. These include working with parents and guardians, as well as with children, in situations of conflict or disability. On the other hand, they refer to weaknesses in disciplinary and investigative training and a lack of any approach to the administrative and legal issues specific to the profession. They recognize, in a negative way, the suspension of classes due to strike situations at both universities, which, they think, could have had an impact on them receiving better teacher training. The positive experiences in the subcategory of Undergraduate Disciplines show the relevance that the relationships they established with some professors had on their careers, and the influence that those experiences had in the construction of their professional identity.

In the subcategory referring to the professional training course, the positive experiences reported by the teachers allude to the reception and guidance provided both by the advising teacher of the educational establishment and by the advising professor from the university. In both cases, the support and learning received are valued as is the possibility

of having been in diverse educational contexts. These could have influenced the decision-making involved in the choice of the type of educational establishment for the exercise of this job. In relation to these experiences, which allude to the treatment received by the advising teachers and the training classes observed, the teachers realize that they were not inspiring. Finally, from the voices of the beginning teachers, recommendations can be drawn to strengthen the university training courses. Within them, systematized support through the mentoring role stands out. This must be in line with the university to allow for the development of leadership skills that allow for the better exercise of the profession.

The findings obtained in this subcategory are related to the studies of Cantón and Tardiff [19], where the relationship between professional training and the construction of the teaching identity is highlighted. These authors point out the positive characteristics of formation during the training course to the link with reality; on the other hand, the negative experiences are consistent with the study carried out by Falcón and Arraiz [23], where the importance of support is pointed out, which addresses emotional support functions among others.

## 6. Conclusions

The purpose of this study was to understand the meaning of the experiences in the trajectory of primary education teachers, who reflect on the construction of their professional identity and their initial training. The study based its conclusions on the findings emanating from the information from semi-structured interviews, in connection with the revised theory. The results of this study confirm the close relationship between the initial training and the construction of the professional identity of the teachers interviewed, who value the experiences and learning obtained in their houses of study, both in the acquisition of disciplinary knowledge as an undergraduate and in the development of their identity and being a teacher, highlighting the ethical component, commitment, social responsibility, and awareness of the need for preparation and permanent updating, all of which coincides with what is indicated by the theory about the construction of professional identity and its multiple edges associated with life experiences within a professional teaching career; this is a permanent process, articulated by the reflection of the teachers' own practices in a framework constituted by affective as well as cognitive, personal, and social dimensions. The results suggest that the process of the construction of the identity of being a teacher is intertwined with the experiences of professional practice, stemming from the initial training in diverse contexts, which influence a better positioning by teachers in the school system, in the definition and understanding of their role, and in their projection as an education professional. These findings support the arguments of the theory that points out the relevance of practices as the first approach to the profession, in real experiences that allow for understanding the dimension of the work environment in which they will develop their pedagogical action. Although the virtues of the professional practices of initial training are recognized, this study also reveals the weaknesses that both groups of qualified teachers from two different universities relate to the preparation they received to perform and assume the challenges of the world of work. Some examples are knowledge about administrative aspects, interaction with parents, and disciplinary and academic management with diverse students. These findings are consistent with what the theory indicates regarding the demands of today's society and the demands of the teaching task, a scenario that becomes complex in the face of requirements ranging from administrative tasks to compliance with national and international measurement standards as well as to providing emotional support to students and families; all of these could be incorporated into guided and articulated professional practices, both by the professor of the subject at the university and by the guiding professor, with a mentoring role, in order to contribute to a robust initial training, as indicated in the literature.

**Author Contributions:** Conceptualization, M.C.-F., E.D.-D. and T.M.-N.; methodology, M.C.-F. and T.M.-N.; software, J.J.-N.; validation, M.C.-F., T.M.-N. and E.D.-D.; formal analysis, J.J.-N.; investigation, M.C.-F.; data curation, M.C.-F.; writing—original draft preparation, M.C.-F. and E.D.-D.; writing—review and editing, J.J.-N.; visualization, T.M.-N.; supervision. All authors have read and agreed to the published version of the manuscript.

**Funding:** This research received no external funding.

**Institutional Review Board Statement:** The study was conducted in accordance with the Declaration of Helsinki and was approved by the Ethics Committee of the University of Santiago de Chile, with protocol code 021.2022 and date of approval 19 January 2022.

**Informed Consent Statement:** Informed consent was obtained from all subjects involved in the study.

**Data Availability Statement:** Not applicable.

**Acknowledgments:** The authors acknowledge the support Doctoral Program in Education, Metropolitan University of Educational Sciences, Chile.

**Conflicts of Interest:** The authors declare no conflict of interest.

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
