# Peer review of "Construction of Teacher Professional Identity through Initial Training"

_education, doi:10.3390/educsci12110822_

Round 1

Reviewer 1 Report

Overall, this is a well organised paper. There is a strong review of the literature and theoretical framework presented. The methodology and methods are clear. The findings are presented to show data excerpts (participant words). And the discussion summarises the main themes that came from the data. I think a bit more could be added to the conclusion to link back to the research question and theoretical framework/lens by which this research was conducted. I wanted a bit more of the "so what?" from this research. How will this impact ITT and the current research?

Author Response

Thank you very much for the observations and comments on our work. We respond to the statements by improving the conclusions and the articulations that occur in the contents that are developed within the investigation. We have attached a second version of the paper for your review.

Reviewer 2 Report

The article intends to use interviews to support the construction of teacher identity based on their initial training.

This research is necessary because the social profile and the profile of students is changing, so that establishing teaching models is essential to adapt to each situation.

The articles to date are based on identifying the patterns that certain groups of teachers have, but there is little literature on the construction of teacher identity based on their initial training.

The presentation of results with their methodology should be improved. Although the questions from the different interviews are presented, it would be convenient and necessary to show a table of results showing the percentage, mean and/or standard deviation of the different questions asked.

Although the study is qualitative, this only refers to data collection and interpretation, which does not preclude the use of quantitative or semi-quantitative techniques in order to draw significant conclusions.

Although the results and discussion are presented in a correct way and there are no inconsistencies, it is necessary to highlight each of the objectives and try to reflect by means of a diagram what are the possible teaching identities constructed or detected from these results.

Author Response

(The authors gave the same response as above.)
